# Generation of DC, AC, and Second-Harmonic Spin Currents by Electromagnetic Fields in an Inversion-Asymmetric Antiferromagnet

**Tatsuhiko N. Ikeda** 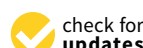

Institute for Solid State Physics, University of Tokyo, Kashiwa, Chiba 277-8581, Japan; tikeda@issp.u-tokyo.ac.jp

**Abstract:** Manipulating spin currents in magnetic insulators is a key technology in spintronics. We theoretically study a simple inversion-asymmetric model of quantum antiferromagnets, where both the exchange interaction and the magnetic field are staggered. We calculate spin currents generated by external electric and magnetic fields by using a quantum master equation. We show that an ac electric field with amplitude $E_0$ leads, through exchange-interaction modulation, to the dc and second-order harmonic spin currents proportional to $E_0^2$. We also show that dc and ac staggered magnetic fields $B_0$ generate the dc and ac spin currents proportional to $B_0$, respectively. We elucidate the mechanism by an exactly solvable model, and thereby propose the ways of spin current manipulation by electromagnetic fields.

**Keywords:** spintronics; spin current; quantum spin system; light-matter interaction; nonlinear optics; quantum master equation; Jordan-Wigner transformation

## 1. Introduction

Spintronics has attracted growing attention in fundamental and applied physics for decades [1–3], where the researchers have explored how to manipulate the spin degree of freedom in materials and devices [4]. For example, the spin Hall effect deriving from the spin-orbit coupling enables the conversion between the spin current and the electric current [5–7], and the spin Seebeck effect [8] extends to the research field of spin caloritronics [9]. One important class of materials in spintronics is the magnetic insulator, where the charge degree of freedom is frozen and magnetic excitations play the principal role [10]. Being free from Ohmic losses, the spin currents in these materials are expected to be useful for future computing devices [11]. Thus, it has been of crucial importance to develop the ways to control these spin currents freely [12].

Antiferromagnets have emerged as a new class of materials whose unique features have turned out to be suited for spintronic applications [13]. For example, the time scale of magnetic excitations of antiferromagnets is typically shorter than that of ferromagnets, the antiferromagnets are promising candidates for high-speed spintronic devices [14]. Among several approaches including thermal effects [15–17], the optical control of antiferromagnets, which enables the fastest manipulation, has attracted considerable attention [18–22]. Recently, Ishizuka and Sato [23,24] theoretically showed that inversion-asymmetric antiferromagnets are useful for spin–current generation by electromagnetic waves. They proposed the spin–current rectification in ac electric and magnetic fields, where the magnitude of the generated dc spin current is proportional to the second power of the input-field amplitude. The dc spin–current generation as rectification has been also numerically confirmed and the second-order harmonic spin current is studied in Ref. [25].

In this paper, we propose two other ways to produce spin currents by electromagnetic fields in inversion-asymmetric antiferromagnets. We consider a one-dimensional model for them, where both

the exchange interaction and the magnetic field are staggered, and study the spin current induced by an electric or magnetic field of pulse shape by numerically integrating a quantum master equation. On the one hand, we show that an ac electric field of amplitude $E_0$ leads to exchange-interaction modulation [26] and gives rise to the dc and second-order harmonic spin currents whose magnitude are proportional to $E_0^2$. Note that this type of coupling between the spin system and the electric field is generic, and thus not restricted to multiferroic systems [23–25]. On the other hand, we show that dc and ac staggered magnetic fields of amplitude $B_0$ generate the dc and ac spin currents, respectively, whose magnitude are both proportional to $B_0$. The underlying mechanism of these spin–current generations are elucidated in a unified manner as the competition between the staggered exchange interaction and magnetic field. This mechanism is distinct from the spin–current rectification proposed in Refs. [23,24].

## 2. Formulation of the Problem

### 2.1. Time-Independent Hamiltonian

In this work, we consider the following Hamiltonian for a spin chain [23]

$$\hat{H}_0 = \sum_{j=1}^{2L} \left\{ J \left[ 1 + (-1)^j \eta_{\text{stag}} \right] (\hat{S}_j^x \hat{S}_{j+1}^x + \hat{S}_j^y \hat{S}_{j+1}^y) + (-1)^j H_{\text{stag}} \hat{S}_j^z \right\}. \tag{1}$$

Here, $\hat{S}_j^\alpha$ ($\alpha = x, y$ and $z$) denote the spin operators at site $j$ for the spin-1/2 representation, $J\,(>0)$ is the exchange interaction, and $\eta_{\text{stag}}$ ($H_{\text{stag}}$) is the staggered exchange interaction (magnetic field). This model, schematically shown in Figure 1, is useful to study inversion-asymmetric antiferromagnets (see, e.g., [25] and references therein for the candidate materials). We impose the periodic boundary conditions $\hat{S}_{2L+j}^\alpha = \hat{S}_j^\alpha$.

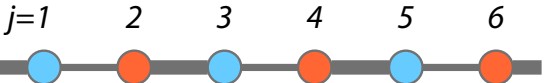

**Figure 1.** Schematic illustration of our model (Equation (1)).The thick and thin bonds represent the exchange couplings $J(1 + \eta_{\text{stag}})$ and $J(1 - \eta_{\text{stag}})$, respectively, and the red and blue sites represent the local magnetic fields $H_{\text{stag}}$ and $-H_{\text{stag}}$, respectively.

There are two kinds of inversion transformation regarding this model: the site-center inversion $\hat{\mathcal{I}}_s$ and the bond-center inversion $\hat{\mathcal{I}}_b$. These inversions are characterized, for instance, by $\hat{\mathcal{I}}_s \hat{S}_j^\alpha \hat{\mathcal{I}}_s^\dagger = \hat{S}_{-j}^\alpha$ and $\hat{\mathcal{I}}_b \hat{S}_j^\alpha \hat{\mathcal{I}}_b^\dagger = \hat{S}_{-j+1}^\alpha$. It follows from Equation (1) that

$$\hat{\mathcal{I}}_s \hat{H}_0(\eta_{\text{stag}}, H_{\text{stag}}) \hat{\mathcal{I}}_s^\dagger = \hat{H}_0(-\eta_{\text{stag}}, H_{\text{stag}}), \qquad \hat{\mathcal{I}}_b \hat{H}_0(\eta_{\text{stag}}, H_{\text{stag}}) \hat{\mathcal{I}}_b^\dagger = \hat{H}_0(\eta_{\text{stag}}, -H_{\text{stag}}). \tag{2}$$

Thus, our Hamiltonian is symmetric under the site-center inversion for $\eta_{\text{stag}} = 0$ and under the bond-center inversion for $H_{\text{stag}} = 0$. When neither $\eta_{\text{stag}}$ nor $H_{\text{stag}}$ vanishes, our Hamiltonian is inversion-asymmetric. As we show below, the spin current arises only for the inversion-asymmetric situation in our setup.

### 2.2. Coupling to AC Electric Field: Difference- and Sum-Frequency Mechanisms

We suppose that our spin model is a low-energy effective model of strongly correlated electrons. Specifically, we regard the exchange interaction $J$ as a superexchange of the one-dimensional Hubbard model at half filling with transfer integral $t_0$ and on-site Coulomb interaction $U$. Then, we obtain $J = 4t_0^2/U$ [27].

Now, we consider the effect of an ac *electric* field along the spin chain. Although the spin chain apparently does not couple to the electric field, it does through virtual hopping processes of the underlying charge degrees of freedom in the Hubbard model [26]. As shown in Refs. [28,29], the ac electric field makes the exchange interaction $J$ be time-dependent as

$$J(t) = \sum_{m,n} (-1)^m \frac{4t_0^2 \mathcal{J}_{n+m}(F) \mathcal{J}_{n-m}(F)}{U - (n+m)\Omega} \cos(2m\Omega t), \tag{3}$$

where $\mathcal{J}_n(x)$ is the Bessel function of the first kind and $\Omega$ the angular frequency of the ac electric field. The dimensionless parameter $F = eaE_0/\Omega$ ($\hbar = 1$ throughout this paper) represents the coupling strength between the electron and the ac electric field, where $e$ ($> 0$) is the elementary charge, $a$ is the lattice constant, and $E_0$ is the field amplitude.

Let us assume that $F \ll 1$ and simplify Equation (3). Under this condition, we have $\mathcal{J}_{n+m}(F)\mathcal{J}_{n-m}(F) = O(F^{|n+m|+|n-m|})$, which implies that the coefficient of $\cos(2m\Omega t)$ is $O(F^m)$ and the higher frequency component rapidly decreases. Thus, we ignore the terms with $|m| \geq 2$ in Equation (3), obtaining

$$J(t) \simeq J + J'(t), \tag{4}$$

$$J'(t) = J\frac{F^2}{2}\frac{\bar{\Omega}^2}{1 - \bar{\Omega}^2} - J\frac{F^2}{2}\frac{\bar{\Omega}^2(1 + 2\bar{\Omega}^2)}{(1 - \bar{\Omega}^2)(1 - 4\bar{\Omega}^2)} \cos(2\Omega t), \tag{5}$$

where $\bar{\Omega} \equiv \Omega/U$ and we ignore the higher-order correction terms in $F$. We assume $\bar{\Omega} < 1$ throughout this paper, and further simplify Equation (5) as

$$J'(t) \simeq J\frac{F^2}{2}\bar{\Omega}^2 [1 - \cos(2\Omega t)] = J\alpha \sin^2(\Omega t) \tag{6}$$

with

$$\alpha \equiv F^2 \bar{\Omega}^2 = \left(\frac{eaE_0}{U}\right)^2. \tag{7}$$

Here, we ignore higher-order correction of $O(\bar{\Omega}^4)$.

We emphasize that the frequencies involved in the exchange interaction in Equation (6) are $0\Omega$ and $2\Omega$ rather than $\Omega$ of the applied ac field. These are kinds of difference-frequency $(\Omega - \Omega)$ and sum-frequency $(\Omega + \Omega)$ generation. The exchange interaction modulation in the spin model derives from the second-order virtual processes of the underlying charge degrees of freedom. In fact, the amplitude $\alpha$ of the exchange interaction modulation is proportional to $E_0^2$ as in Equation (7). Thus, as we show below, the linear response as a spin model to the exchange interaction modulation $\alpha$ gives rise to the dc ($0\Omega$) and second-order harmonic ($2\Omega$) outputs.

### 2.3. Total Hamiltonian and Spin Current

We complete the formulation of the problem that we address in this paper. Combining the above arguments, we arrive at the following spin-system Hamiltonian

$$\hat{H}_{\text{tot}}(t) = \hat{H}_0 + \hat{H}_{\text{ext}}(t), \tag{8}$$

$$\hat{H}_{\text{ext}}(t) = J'(t) \sum_{j=1}^{2L} \left[1 + (-1)^j \eta_{\text{stag}}\right] (\hat{S}_j^x \hat{S}_{j+1}^x + \hat{S}_j^y \hat{S}_{j+1}^y), \tag{9}$$

where $\hat{H}_0$ is defined in Equation (1).

Note that the total Hamiltonian $\hat{H}_{\text{tot}}(t)$ has the global U(1) symmetry associated with the rotation around the $S^z$ axis. Thus, the total magnetization $\hat{M}^z \equiv \sum_j \hat{S}_j^z/2$ is conserved, and the

continuity equations for local $\hat{S}_i^z$s hold true: $d\hat{S}_i^z/dt + \hat{j}_i^{\text{spin}} - \hat{j}_{i-1}^{\text{spin}} = 0$. Here, $\hat{j}_j^{\text{spin}} \equiv [J + J'(t)][1 + (-1)^j \eta_{\text{stag}}](\hat{S}_j^x \hat{S}_{j+1}^y - \hat{S}_j^y \hat{S}_{j+1}^x)$ represents the local spin current flowing between the sites $j$ and $j+1$. The observable of interest is the total spin current

$$\hat{J}_{\text{spin}} = \sum_j \hat{j}_j^{\text{spin}} = [J + J'(t)]\sum_j [1 + (-1)^j \eta_{\text{stag}}](\hat{S}_j^x \hat{S}_{j+1}^y - \hat{S}_j^y \hat{S}_{j+1}^x). \tag{10}$$

Since we focus on the case in which $J'(t)$ is small, we may safely neglect $J'(t)$ from Equation (10). In the following, we consider the ground state of $\hat{H}_0$ and analyze the spin current $\hat{J}_{\text{spin}}$ generated by the time-dependent perturbation $\hat{H}_{\text{ext}}(t)$.

Note that the spin current is parallel to the ac electric field, which has been assumed to be along the chain. If we applied the ac electric field perpendicular to the chain, the exchange interaction modulation would not happen and there would be no spin current generation in our one-dimensional model. This is not true in general for two-dimensional systems. In fact, Naka et al. recently showed that a dc electric field or thermal gradient leads to a spin current perpendicular to it in two-dimensional organic antiferromagnets [17].

For later use, we remark that $\hat{J}_{\text{spin}}$ is odd under both inversions:

$$\hat{\mathcal{I}}_s \hat{J}_{\text{spin}} \hat{\mathcal{I}}_s^\dagger = \hat{\mathcal{I}}_b \hat{S} \hat{\mathcal{I}}_b^\dagger = -\hat{J}_{\text{spin}}. \tag{11}$$

Therefore, when $\eta_{\text{stag}} = 0$ or $H_{\text{stag}} = 0$ and either inversion symmetry is present, no dynamics occurs in the spin current. For the spin-current generation to occur, both $\eta_{\text{stag}}$ and $H_{\text{stag}}$ must be nonzero.

## 3. Results

### 3.1. DC and Second-Harmonic Spin Currents

We now numerically investigate the spin current dynamics under a multi-cycle pulse field of experimental interest. We replace $J'(t)$ in Equation (6) by

$$J'(t) \rightarrow J'_{\text{pulse}}(t) \equiv J\alpha f(t) \sin^2(\Omega t), \tag{12}$$

where the Gaussian envelope function $f(t) = \exp[-4\ln 2 \cdot (t/T_{\text{FWHM}})^2]$ with full width at half maximum $T_{\text{FWHM}}$. To be specific, we set $T_{\text{FWHM}} = 10\pi/\Omega$, for which $J'_{\text{pulse}}(t)$ is illustrated in Figure 2a. We confirmed that the results are not sensitive to the pulse width.

The actual numerical calculations are as follows. We take an initial time $t_{\text{ini}}$ ($< 0$) so that $J'_{\text{pulse}}(t_{\text{ini}})$ is negligibly small, and suppose that the spin system is in the ground state with zero total magnetization $\hat{M}^z = \sum_j \hat{S}_j^z/2 = 0$. Then, we solve the dynamics represented by a quantum master equation (see Section 5 for detail), which describes the time-dependent Schrödinger equation in the presence of relaxation. We set the relaxation rate as $\gamma = 0.1J$. Our master equation ensures that, without the external field, the system relaxes to the ground state, i.e., the zero-temperature state. Thus, thermal fluctuations [15] are neglected in our model.

Figure 2b shows a typical time profile of the spin current. The Hamiltonian parameters are $(\eta_{\text{stag}}, H_{\text{stag}}/J) = (0.1, 0.03)$ and the field parameters are $\Omega = 5 \times 10^{-2}J$ and $\alpha = 0.1$. Figure 2c shows the corresponding Fourier spectrum $|J_{\text{spin}}(\omega)|$, which consists of the dc ($\omega = 0$), second-order harmonic ($\omega = 2\Omega$), and fourth-order harmonic ($\omega = 4\Omega$) components. As emphasized in Section 2, there appear even-order harmonics because the exchange-interaction modulation in Equation (12) consists of the dc and second-order harmonic components and no longer involves the fundamental frequency $\Omega$ of the input laser.

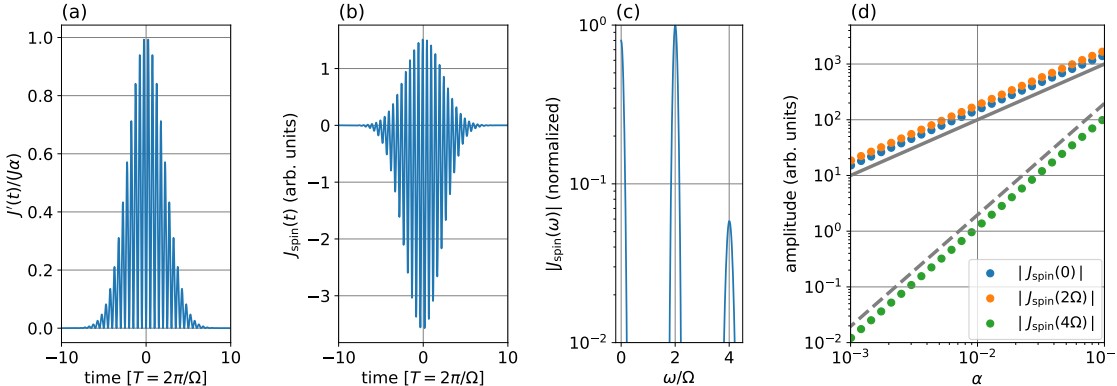

**Figure 2.** (**a**) Form of oscillating exchange interaction $J'_{\text{pulse}}$ (12). (**b**) Time profile of spin current $J_{\text{spin}}(t)$ for $(\eta_{\text{stag}}, H_{\text{stag}}/J) = (0.1, 0.03)$ and $\Omega = 5 \times 10^{-2} J$ and $\alpha = 0.1$. (**c**) Amplitude of the corresponding Fourier transform. (**d**) Amplitudes of the dc and harmonic spin currents, $|J_{\text{spin}}(n\Omega)|$ ($n = 0, 2$, and $4$), plotted against the exchange-interaction-modulation amplitude $\alpha$. The solid (dashed) line shows the slope 1 (2) for the guide to the eye.

The laser-intensity dependence of each harmonic spin current is shown in Figure 2d. In the log-log scale, the dc and second-order harmonic data follow a line with slope 1, whereas the fourth-order harmonic ones with slope 2. Therefore, the dc and second-order harmonic components are proportional to $\alpha$. Meanwhile, the fourth-order harmonic component is proportional to $\alpha^2$ and, thus, arises from the second-order process in terms of $\alpha$. In terms of the ac-electric-field amplitude $E_0$, the dc and second-order harmonic spin currents are $O(E_0^2)$ whereas the fourth-order harmonic is $O(E_0^4)$. Note that the fourth-order harmonic spin current may become different if we incorporate the terms with $m = \pm 2$ in Equation (3), which are $O(E_0^4)$ and neglected in our calculation. However, the dc and second-order harmonic spin currents are not affected much by these higher-order terms.

## 3.2. Direction of DC Spin Current

In Section 3.1, we show a typical behavior of the spin current by fixing $(\eta_{\text{stag}}, H_{\text{stag}}/J) = (0.1, 0.03)$. Here, we focus on the dc component and study how its direction depends on the Hamiltonian parameters $\eta_{\text{stag}}$ and $H_{\text{stag}}$.

Figure 3 shows the dc spin current in the $(\eta_{\text{stag}}, H_{\text{stag}})$-plane obtained by a similar calculation as in Section 3.1. The color bar is renormalized by a positive scale factor for display. Since we work in the linear response regime, the color map does not change with variation of $\alpha$ as long as $\alpha$ is sufficiently small. In the first quadrant $\eta_{\text{stag}} > 0$ and $H_{\text{stag}} > 0$, the dc spin current is negative. Note that this is consistent with Figure 2b, where the dc component, or the time average of $J_{\text{spin}}(t)$, is negative.

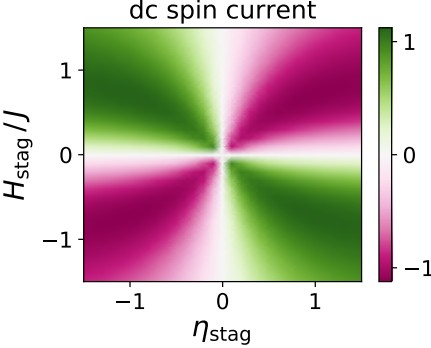

**Figure 3.** Rescaled dc spin current calculated in the pulse dynamics over the $(\eta_{\text{stag}}, H_{\text{stag}})$-plane. The other parameters are set as $T_{\text{FWHM}} = 10\pi/\Omega$, $\Omega = 5 \times 10^{-2} J$ and $\alpha = 0.1$.

The dc component vanishes on the lines $\eta_{\text{stag}} = 0$ and $H_{\text{stag}} = 0$ and its magnitude increases as $(\eta_{\text{stag}}, H_{\text{stag}})$ goes away from these lines. On these lines, either the bond-center or site-center inversion symmetry arises, and not only the dc component but also the total spin current $J_{\text{spin}}(t)$ vanishes, as shown in Section 2.

This inversion-symmetry argument explains why Figure 3 is antisymmetric under reflections across each of the $\eta_{\text{stag}}$ and $H_{\text{stag}}$ axes. As shown in Equation (2), the sign change of $H_{\text{stag}}$ ($\eta_{\text{stag}}$) is equivalent to applying the site-center (bond-center) inversion. On the other hand, each of the site-center and bond-center inversion changes the sign of the spin current as in Equation (11). From these two properties, it follows that

$$J_{\text{spin}}^{-\eta_{\text{stag}}, H_{\text{stag}}}(t) = J_{\text{spin}}^{\eta_{\text{stag}}, -H_{\text{stag}}}(t) = -J_{\text{spin}}^{\eta_{\text{stag}}, H_{\text{stag}}}(t), \tag{13}$$

and, hence, similar relations hold true for the dc components.

### 3.3. Mechanism of Spin Current Generation

Here, we look into the mechanism of the spin current generation numerically obtained in the previous sections. For this purpose, we focus on the special case of $\eta_{\text{stag}} = 1$, for which the exchange interaction vanishes at every two bonds and the spin chain is dimerized as illustrated in Figure 4a. Thus, our spin chain problem reduces to a single dimer.

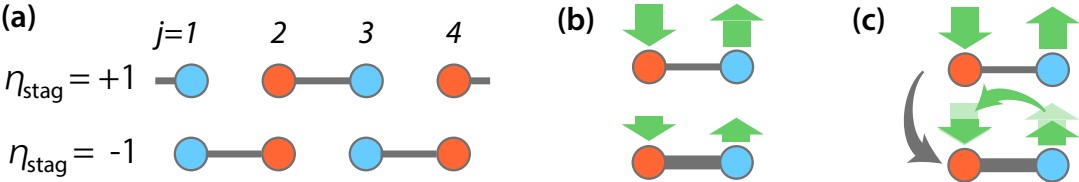

**Figure 4.** (**a**) Schematic illustration of our model (1) for $\eta_{\text{stag}} = 1$ (top) and $\eta_{\text{stag}} = -1$ (bottom). The missing bonds show that the exchange couplings vanish on them. (**b**) Isolated dimer in Equation (14) for $H_{\text{stag}} > 0$, where the red (blue) site shows the positive (negative) local magnetic field. The green arrows represent the local magnetization on each site in the ground state in Equation (16). The top (bottom) plot shows the dimer for the weaker (stronger) exchange interaction. (**c**) Mechanism of spin current generation by exchange-interaction modulation. As the exchange interaction increases, the local magnetization flows from right to left when $H_{\text{stag}} > 0$.

To investigate the ground state of a dimer, we take $j = 2$ and 3, for example. Writing the spin operators as $\hat{S}_L^\alpha = \hat{S}_2^\alpha$ and $\hat{S}_R^\alpha = \hat{S}_3^\alpha$ ($\alpha = x, y$, and $z$), we have the following Hamiltonian:

$$\hat{H}_{\text{dimer}} = 2J(\hat{S}_L^x \hat{S}_R^x + \hat{S}_L^y \hat{S}_R^y) + H_{\text{stag}} \hat{S}_L^z - H_{\text{stag}} \hat{S}_R^z. \tag{14}$$

Among the four states $|\uparrow\uparrow\rangle$, $|\uparrow\downarrow\rangle$, $|\downarrow\uparrow\rangle$, and $|\downarrow\downarrow\rangle$, each of $|\uparrow\uparrow\rangle$ and $|\downarrow\downarrow\rangle$ is the eigenstate of $\hat{H}_{\text{dimer}}$ with zero eigenvalue. On the other hand, $|\uparrow\downarrow\rangle$ and $|\downarrow\uparrow\rangle$ couple to each other. In the basis of these states, $\hat{H}_{\text{dimer}}$ is represented by the $2 \times 2$ matrix:

$$H_{\text{dimer}} = \begin{pmatrix} H_{\text{stag}} & J \\ J & -H_{\text{stag}} \end{pmatrix}, \tag{15}$$

whose eigenvalues are $\pm\sqrt{J^2 + H_{\text{stag}}^2}$. Thus, the ground state $|\text{gs}\rangle$ is the eigenstate with negative eigenvalue and given by

$$|\text{gs}\rangle = -\sin\frac{\theta}{2}|\uparrow\downarrow\rangle + \cos\frac{\theta}{2}|\downarrow\uparrow\rangle; \qquad \cos\theta = \frac{H_{\text{stag}}}{\sqrt{J^2 + H_{\text{stag}}^2}}. \tag{16}$$

The local magnetization distribution in $|\text{gs}\rangle$ follows from the exact solution:

$$\langle\text{gs}|\hat{S}_L^z|\text{gs}\rangle = -\frac{H_{\text{stag}}}{2\sqrt{J^2 + H_{\text{stag}}^2}} = -\langle\text{gs}|\hat{S}_R^z|\text{gs}\rangle. \tag{17}$$

For the special case of $H_{\text{stag}} = 0$, the local magnetization on both sites vanish and there is no magnetization imbalance between the sites. This is a manifestation of the bond-center inversion symmetry. For $H_{\text{stag}} > 0$, the local magnetization on the left (right) is negative (positive), and this situation is illustrated in Figure 4b. These signs become opposite for $H_{\text{stag}} < 0$.

As the exchange interaction $J$ increases, the magnetization imbalance between the sites becomes smaller, as illustrated in Figure 4b. This tendency can be read from Equation (17). In addition, we can make an intuitive interpretation as follows. In the limit of $J \to 0$, there is no spin exchange and $|\text{gs}\rangle = |\downarrow\uparrow\rangle$ for $H_{\text{stag}} > 0$ so that the system maximizes each local magnetization and minimizes the energy. As $J$ is turned on, the system decreases its energy further by using the spin exchange, where the local magnetizations are decreased. In fact, in the limit $J \to \infty$, the ground state becomes the spin singlet pair, which has no local magnetization. There exists a competing effect between $J$ and $H_{\text{stag}}$: $J$ prefers the spin singlet and less local magnetizations whereas $H_{\text{stag}}$ does larger magnetization imbalance.

From the above argument, we arrive at understanding the spin current generation. For $H_{\text{stag}} > 0$, the increase of $J$ corresponds to the transition from the top picture to the bottom one in Figure 4b. Here, the magnetization flows from right to left in total, or the dc spin current is negative, as illustrated in Figure 4c. This explains why the first quadrant of Figure 3 presents negative values. Note that the continuity equation for magnetization does not hold true exactly, unlike that for electric charge, owing to the dissipation, as shown in Section 5.

It is now clear that the dc spin current is positive for $\eta_{\text{stag}} < 0$. In this case, the signs of local magnetic fields and, hence, local magnetic moments become opposite in Figure 4b, and the direction of the dc spin current is thus flipped. Furthermore, it is also clear that the dc spin current changes its sign if $\eta_{\text{stag}}$ is changed from $+1$ to $-1$. This change leads to the other parings of dimers, as shown in Figure 4a (bottom). Here, the local magnetic field on the left and right sites of the dimer is flipped and, thus, the dc spin current changes its sign.

These interpretations are basically true for the general case $\eta_{\text{stag}} \neq \pm 1$. Unless $\eta_{\text{stag}} = 0$, the exchange interaction is alternating. Focusing on a bond with the stronger exchange, we regard the two sites on the bond forming a dimer. Unless $H_{\text{stag}} = 0$, the local magnetic fields on the two sites are different and some magnetization imbalance exists in the dimer. Then, the increase of exchange interaction decreases the magnetization imbalance and the dc spin current arises accordingly.

### 3.4. DC Spin Current Generation by External Magnetic Field

The spin–current generation mechanism elucidated in the previous section is the competing effect between the exchange couplings and the local magnetic fields. This implies that the spin currents can also be generated by staggered magnetic fields. To show this, we replace the $\hat{H}_{\text{ext}}(t)$ discussed thus far by the following term:

$$\hat{H}'_{\text{ext}}(t) = B(t)\sum_{j=1}^{2L}(-1)^j\hat{S}_j^z, \qquad B(t) \equiv J\beta f(t) \cdot \begin{cases} 1 & \text{(dc case)}, \\ \cos(\Omega t) & \text{(ac case)}, \end{cases} \tag{18}$$

where $f(t)$ is the Gaussian envelope function defined below in Equation (12). For the ac case, we use $\Omega = 5 \times 10^{-2} J$ and $T_{\text{FWHM}} = 10\pi/\Omega = 2\pi \times 10^2 J^{-1}$ as in the previous sections. For the dc case, we use the same pulse width $T_{\text{FWHM}} = 2\pi \times 10^2 J^{-1}$. The forms of $B(t)$ for the dc and ac cases are shown in Figure 5a. The ac case is studied in Refs. [23,25], and we compare the dc case results to it below.

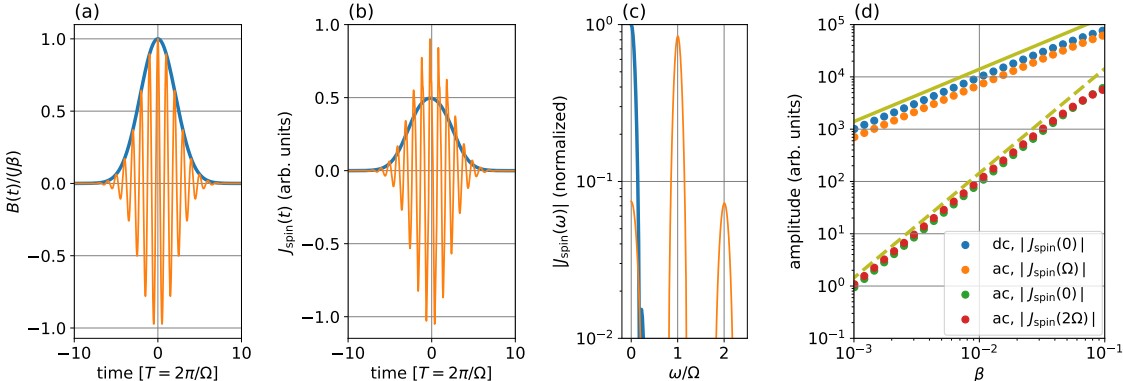

**Figure 5.** (**a**) Form of external magnetic field for the dc (blue) and ac (orange) cases. (**b**) Time profile of spin current $J_{\text{spin}}(t)$ for the dc (blue) and ac (orange) cases. The parameters are $(\eta_{\text{stag}}, H_{\text{stag}}/J) = (0.1, 0.03)$, $\Omega = 5 \times 10^{-2} J$, and $\beta = 0.1$. (**c**) Amplitude of the corresponding Fourier transform for the dc (blue) and ac (orange) cases. The amplitude is normalized so that $|J_{\text{spin}}(\omega = 0)| = 1$ for the dc case. (**d**) Amplitudes of the dc and harmonic spin currents plotted against the magnetic-field amplitude $\beta$. The blue show the dc spin current $|J_{\text{spin}}(0)|$, and the orange, green, and red show $|J_{\text{spin}}(n\Omega)|$ with $n = 1, 0$, and 2), respectively. The solid (dashed) line shows the slope 1 (2) for the guide to the eye.

In the experimental viewpoint, the external magnetic field may be spatially uniform. Since we consider the situation where internal staggered magnetic fields $H_{\text{stag}}$ are present, the external field induces the staggered component (Equation (18)) in general [23]. Note that the uniform component of the external magnetic field causes no physical effect since the total magnetization is a conserved quantity in our model.

Figure 5b shows the time profiles of the spin currents for the dc and ac cases. For the dc case, the generated spin current is positive in contrast to the exchange-interaction modulation in Figure 2b. This is in perfect agreement with the physical mechanism found in Section 3.3 as follows. Let us focus on the dimer limit and look at Figure 4b. Our external magnetic field for the dc case in Equation (18) increases the difference between the local magnetic fields and, hence, the local magnetizations on the left and right sites. This amounts to the transfer of some positive local magnetization from left to right, resulting in the positive spin current.

The corresponding spin current spectra are shown in Figure 5c. For the ac case, there are several harmonic peaks, as discussed by Ikeda and Sato [25]. In particular, the dc component is generated by the spin–current rectification [23]. As shown in Figure 5d, the ac component $|J_{\text{spin}}(\Omega)|$ is proportional to $\beta$, and the dc and second-order harmonic components are to $\beta^2$. In other words, the results of the ac case are understood by perturbation in $\beta$. Whereas the ac output is the linear response, the dc and second-order harmonic ones are the second-order perturbation.

Our finding is that the dc spin current for the dc input is proportional to $\beta$ rather than $\beta^2$, as shown in Figure 5. Thus, this dc spin current is significantly larger for smaller magnetic fields. Again, the mechanism of the spin current generation is the one elucidated in Section 3.3, and one notices that the direction of the dc spin current is reversed by changing the sign of $\beta$. Therefore, the dc-spin–current direction can be switched by changing the direction of the external magnetic field.

## 4. Discussion and Conclusions

Studying a simple model of inversion-asymmetric antiferromagnets, we propose two ways of generating spin currents. The first one is to utilize an ac electric field, which leads, through the exchange-interaction modulation, to the dc and second-order harmonic spin currents. This finding serves as an interesting application of the exchange-interaction control [26]. The amplitude of the generated spin current by this method scales as $E_0^2$ with $E_0$ being the amplitude of the input ac electric field. This second-power scaling in electromagnetic fields are in common with the different proposals including the spin–current rectification proposed in Refs. [23,24]. Thus, the relative importance between these proposals relies on the prefactors that should depend on the material parameters.

The second way of spin–current generation is to utilize a dc magnetic field of pulse shape. In this case, the generated spin current is proportional to the amplitude of the external magnetic field. This scaling is better for generating larger spin currents than the second-power scaling proposed in related studies [23–25]. In addition, the direction of the spin current can be reversed by changing the direction of the external magnetic field. This controllability could be of experimental relevance.

Both ways of spin current generation are understood in a unified manner as in Section 3.3. In inversion-asymmetric antiferromagnets, there exists some imbalance of local magnetizations at equilibrium. Once either the exchange interaction or the local magnetic field is modulated, the magnetization imbalance is converted into spin currents. We note that this is a transient phenomenon. In fact, instead of the pulse, we could turn on the exchange-interaction modulation and keep it constant for a very long time. The dc spin current in this situation [29] would decrease as the system approaches a steady state.

The spin current generation mechanism proposed in this paper is very simple and generic. This mechanism should contribute to the understanding of spin currents in antiferromagnets, and its experimental verification is of interest in fundamental and applied physics. Upon experimental verifications, one might need more material-specific models including crystallography and so on [30]. This future direction is of crucial importance in applications.

## 5. Materials and Methods

### 5.1. Fermionization

In actual calculations, it is convenient to map our spin model in Equations (1), (8) and (9) onto noninteracting spinless fermions. Following Ikeda and Sato [25], we perform the Jordan–Wigner transformation [31]: $\hat{S}_j^+ = \prod_{i(<j)}(1 - 2\hat{c}_i^\dagger \hat{c}_i)\hat{c}_j$, $\hat{S}_j^- = \prod_{i(<j)}(1 - 2\hat{c}_i^\dagger \hat{c}_i)\hat{c}_j^\dagger$, and $\hat{S}_j^z = 1/2 - \hat{c}_j^\dagger \hat{c}_j$, where $\hat{S}_j^\pm \equiv (\hat{S}_j^x \pm i\hat{S}_j^y)/2$ and the creation and annihilation operators satisfy the standard anticommutation relations $\{\hat{c}_i, \hat{c}_j^\dagger\} = \delta_{ij}$, etc.

Then, we simplify our spin model further by defining the Fourier transforms for the odd and even sites: $\hat{a}_k \equiv L^{-1/2}\sum_{j=1}^{L} e^{-ik(2j)}\hat{c}_{2j}$ and $\hat{b}_k \equiv L^{-1/2}\sum_{j=1}^{L} e^{-ik(2j+1)}\hat{c}_{2j+1}$, where $k = \pi m/L$ ($m = 0, 1, \ldots, L-1$). The spin Hamiltonians are then mapped to $2 \times 2$ matrices with the following two-component fermion operator: $\psi_k \equiv {}^t(\hat{a}_k, \hat{b}_k)$. In fact, one obtains

$$\hat{H}_0 = \sum_k \psi_k^\dagger H_0(k)\psi_k; \qquad H_0(k) = J\left(\cos k\,\sigma_x - \eta_{\text{stag}}\sin k\,\sigma_y\right) - H_{\text{stag}}\sigma_z, \qquad (19)$$

$$\hat{H}_{\text{ext}}(t) = \sum_k \psi_k^\dagger H_{\text{ext}}(k,t)\psi_k; \qquad H_{\text{ext}}(k,t) = J'(t)\left(\cos k\,\sigma_x - \eta_{\text{stag}}\sin k\,\sigma_y\right), \qquad (20)$$

where $\sigma_\alpha$ ($\alpha = x, y,$ and $z$) are the Pauli matrices. The spin current in Equation (10) is also fermionized as

$$\hat{J}_{\text{spin}} = \sum_k \psi_k^\dagger J_{\text{spin}}(k)\psi_k; \qquad J_{\text{spin}}(k) = -J(\sin k\,\sigma_x + \eta_{\text{stag}}\cos k\,\sigma_y). \qquad (21)$$

We remark that the Hamiltonian and the spin current are represented as sums over $k$. Thus, our problem reduces to the direct product of each $k$-subspace.

We let $|\phi_{\pm}(k)\rangle$ denote the two eigenstates of $H_0(k)$,

$$H_0(k)\,|\phi_{\pm}(k)\rangle = \pm\epsilon(k)\,|\phi_{\pm}(k)\rangle \tag{22}$$

with $\epsilon(k) = \sqrt{J^2\left[\cos^2 k + (\eta_{\text{stag}}\sin k)^2\right] + H_{\text{stag}}^2}$. These eigenvalues $\pm\epsilon(k)$ define the two energy bands, which are illustrated in Figure 6. The band gap is given by $\Delta = 2\sqrt{J^2\min(1,\eta_{\text{stag}}^2) + H_{\text{stag}}^2}$. The ground state of the total system is such a state that all the states in the lower (upper) band are occupied (unoccupied). Note that this ground state is half-filled and, hence, has zero magnetization in the spin language.

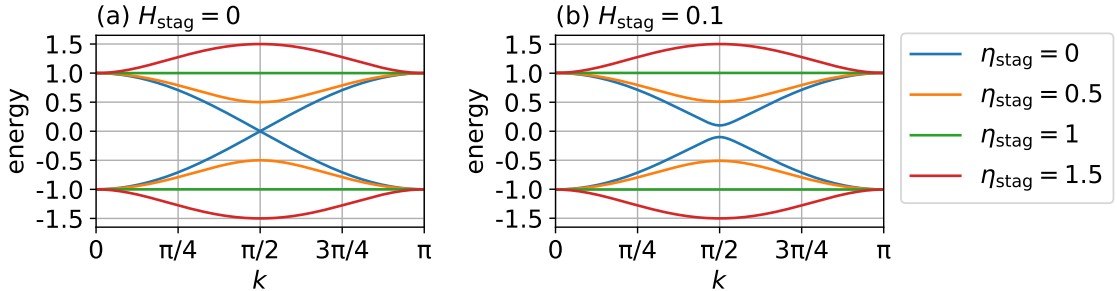

**Figure 6.** Energy bands for four choices of the staggered exchange interaction $\eta_{\text{stag}}$ in the absence (**a**) and presence (**b**) of the staggered magnetic field $H_{\text{stag}}$. Each parameter is shown in the figure, and $J$ is set to unity.

## 5.2. Quantum Master Equation

We have analyzed the dynamics under pulse fields by using the quantum master equation (see [25] for more detail):

$$\frac{\mathrm{d}}{\mathrm{d}t}\rho(k,t) = -\mathrm{i}[H(k,t),\rho(k,t)] + \mathcal{D}[\rho(k,t)], \tag{23}$$

$$\mathcal{D}[\rho(k,t)] \equiv \gamma\left(L_k\rho(k,t)L_k^{\dagger} - \frac{1}{2}\{L_k^{\dagger}L_k,\rho(k,t)\}\right), \tag{24}$$

where $\rho(k,t)$ is the $2\times 2$ reduced density matrix for the $k$-subspace. In the Hamiltonian matrix $H(k,t) = H_0(k) + \hat{H}_{\text{ext}}(k,t)$, the time-dependent exchange interaction is replaced by the pulse one as Equation (12). The first term on the right-hand side of Equation (23) is the same as the time-dependent Schrödinger equation, whereas the second one describes the relaxation effect. The Lindblad operator $L_k \equiv |\phi_{-}(k)\rangle\langle\phi_{+}(k)|$ causes the excited state $|\phi_{+}(k)\rangle$ relaxing to the ground state $|\phi_{-}(k)\rangle$ at rate $\gamma$ [32]. Thus, our model corresponds to the case that the system is in contact with a reservoir at zero temperature, and thermal fluctuations are neglected.

Our master equation (Equation (23)) is a set of ordinary differential equations. Thus, we can numerically solve it by an explicit method such as the Runge–Kutta method.

Finally, we remark that the continuity equation $d\hat{S}_i^z/dt + \hat{j}_i^{\text{spin}} - \hat{j}_{i-1}^{\text{spin}} = 0$ is corrected by the $\mathcal{D}$ term in the master equation. To see this, we consider the full master equation $d\hat{\rho}/dt = -\mathrm{i}[\hat{H}_{\text{tot}}(t),\hat{\rho}] + \hat{\mathcal{D}}(\hat{\rho})$ instead of Equation (23) reduced to each $k$. Then, the expectation value $S_j^z(t) = \mathrm{tr}(\hat{\rho}(t)\hat{S}_j^z)$ satisfies the following equation

$$\frac{\mathrm{d}S_j^z(t)}{\mathrm{d}t} + j_i^{\text{spin}}(t) - j_{i-1}^{\text{spin}}(t) = \mathrm{tr}\{\hat{\mathcal{D}}[\hat{\rho}(t)]\hat{S}_j^z\}. \tag{25}$$

The right-hand side gives the source or sink for the magnetization and, hence, the standard continuity equation does not hold true. Note that this is not a contradiction because the magnetization, or the angular momentum along a certain direction, is not conserved in general unlike the electric charge that is strictly conserved.

**Funding:** This work was funded by JSPS KAKENHI Grant No. JP18K13495.

**Acknowledgments:** Fruitful discussions with K. Chinzei, M. Sato, and H. Tsunetsugu are gratefully acknowledged.

**Conflicts of Interest:** The author declares no conflict of interest.

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
