# Peer review of "Generation of DC, AC, and Second-Harmonic Spin Currents by Electromagnetic Fields in an Inversion-Asymmetric Antiferromagnet"

_condensedmatter, doi:10.3390/condmat4040092_

Round 1

Reviewer 1 Report

It is my pleasure to review the paper “Generation of dc, ac and second-harmonic spin currents by electromagnetic fields in an inversion-asymmetric antiferromagnet” by T. N. Ikeda. The paper is well organised and written with important idea to generate spin currents – one of the very basically crucial issues for spintronic applications. Idea of generation of spin current in antiferromagnets is vital, especially by electric field as it is potential for application in electric field controlled devices working with low power consumption. T. N. Ikeda has theoretically established an approach by using an ac electric field through the exchange-interaction modulation. Together with another idea of generating spin current using a dc magnetic field, I find these ideas so interesting and realistic for practical applications.

I would like to send the author several criticisms:

Thermal fluctuation would importantly affect generation of spin (e.g. Seki et al., Phys Rev Lett 2015, https://doi.org/10.1103/PhysRevLett.115.266601). The author did not discuss how the thermal term was considered. Spin-orbit should also be taken into account (Naka et al., Nature Comm. 2019, https://www.nature.com/articles/s41467-019-12229-y). Could author have any discussion or consideration? Crystallography could also be important to consider especially defects (Rezende et al., J. Appl. Phys. 2019 https://doi.org/10.1063/1.5109132). The author unlikely considered?

The author unlikely discussed about those terms which could importantly effect the generation of spin current. I hope the author would carefully consider those.

The paper would be worth for a publication if my criticisms are addressed in revised manuscript.

Author Response

I thank the referee for taking his/her valuable time to review my manuscript and positively evaluating my work. The comments raised by the referee have helped revise the manuscript. In the revised manuscript, I have added discussions to compare my results and the literature that the referee kindly told me. Below, I address each comment.

1. "Thermal fluctuation would importantly affect generation of spin (e.g. Seki et al., Phys Rev Lett 2015, https://doi.org/10.1103/PhysRevLett.115.266601). The author did not discuss how the thermal term was considered."

My calculation corresponds to the case that the spin system is in contact with a zero-temperature reservoir. In the revised version, I have clearly stated this in Secs. 3.1 and 5.2.

2. "Spin-orbit should also be taken into account (Naka et al., Nature Comm. 2019, https://www.nature.com/articles/s41467-019-12229-y). Could author have any discussion or consideration?"

Thank you for telling me the recent paper by Naka et al., which I have found very relevant to my work. They have shown that the spin-orbit is not necessary. This is a similarity between my work and theirs. There are two differences. (1) I consider a one-dimensional model whereas they two-dimensional one. (2) The spin current is parallel to the electric field in my work whereas perpendicular in their work.

In the revised manuscript, I have made comparisons with their results by citing the paper in Sec. 2.3.

3. "Crystallography could also be important to consider especially defects (Rezende et al., J. Appl. Phys. 2019 https://doi.org/10.1063/1.5109132). The author unlikely considered?"

Yes, it is true that crystallography would be important to study real materials. The aim of my work is showing the basic idea in the simplest way, and I would like to say that detailed studies of various materials are interesting and important future works.

In the revised version, I have remarked the importance of those aspects at the end of Sec.4 with citing the article.

Reviewer 2 Report

The manuscript titled by "Generation of dc, ac, and second-harmonic spin currents by electromagnetic fields in an inversion-asymmetric antiferromagnet", reports a theoretical study of spin currents generated by external electric and magnetic fields in a simple inversion-asymmetric antiferromagnet by using a quantum master equation, thereby propose the ways of spin current manipulation by electromagnetic field.  The model is simple but the results are very interesting. The proposed methods to produce spin current are found to be proportional to E^2 or to B. It will be more helpful that the authors can clarify the relationship between the methods proposed here and experimental approaches.  And most importantly, the author should provide more information about the advantage to manipulate spin currents in a antiferromagnet insulator, and what is the new physics in the model. Due to lack of new physics in this work, I cannot recommend the publication of the paper in its current form. 

Author Response

I thank the referee for reviewing my manuscript and for his/her positive evaluation "The model is simple but the results are very interesting." I also appreciate his/her comments, which have enabled me to emphasize the importance and the novelty in the revised manuscript. Below I list my responses to each of the comments.

1. "It will be more helpful that the authors can clarify the relationship between the methods proposed here and experimental approaches."

Regarding the spin current generation, the existing experimental approaches mostly use the thermal gradient. Compared with this approach, the electromagnetic-field method proposed in this work enables us to realize faster manipulations.

In the revised Introduction, I have pointed out this aspect.

2. "And most importantly, the author should provide more information about the advantage to manipulate spin currents in a antiferromagnet insulator"

One advantage of using magnetic insulators is that there is no Ohmic loss, and the device will be more energy-saving. One advantage of using antiferromagnets is that the time scale of them is typically smaller than that of ferromagnets. Thus the antiferromagnets are advantageous in high-speed device applications.

These points have been emphasized in the Introduction in the revised version.

3. "what is the new physics in the model" and your comment to the editor "The major issue of this manuscript is that the scientific significance of this paper is very similar to other papers, refs. [22,23,24]. In specific, regarding the authors' other paper, Ref. [24] "High-harmonic generation by electric polarization, spin current and magnetization", the authors should clarify the reason why they submit their works based on same theoretical
model to two journals."

There are two new things in the present study.

The first is the coupling between the system and the electromagnetic fields. While the previous studies discuss the multiferroic coupling, the present study considers the exchange-interaction modulation. Since the number of multiferroic materials is quite limited, the mechanism presented in this work should be more generic.

The second is the physical mechanism of the spin current generation. In the present study, the dc spin current arises from the dc component of the coupling at the level of the spin model. On the other hand, in the previous studies, it arises from the second-order perturbation of the coupling. Thus the physical mechanism is very different and the new mechanism in the present study has been explained in a surprisingly transparent way in Sec. 3.3.

Indeed I use a similar numerical technique in the analysis, but the physics that we discuss in this work is significantly different from that in previous studies.

In the revised manuscript, I have emphasized the above differences in the Introduction.

Reviewer 3 Report

The main topic of the manuscript belongs to the emerging field of anti-ferromagnetic materials spintronics. Authors obtained original and new results which should soon be verified experimentally, even if the paper is very theoretical. Also, the text is very well written giving some didactic explainations. I strongly suggest to publish it in the actual shape.

Author Response

I thank the referee for his/her most positive evaluation. I would like to keep up the good work.